# Polarization of Melatonin-Modulated Colostrum Macrophages in the Presence of Breast Tumor Cell Lines

**DOI:** 10.3390/ijms241512400

**Published:** 2023-08-03

**Authors:** Kenia Maria Rezende Silva, Danielle Cristina Honório França, Adriele Ataídes de Queiroz, Danny Laura Gomes Fagundes-Triches, Patrícia Gelli Feres de Marchi, Tassiane Cristina Morais, Adenilda Cristina Honorio-França, Eduardo Luzía França

**Affiliations:** 1Postgraduate Program in Basic and Applied Immunology and Parasitology, Federal University of Mato Grosso, Barra do Garças 78600-000, MT, Braziladrieleaqueiroz@hotmail.com (A.A.d.Q.); dannylauragf@hotmail.com (D.L.G.F.-T.); dr.eduardo.franca@gmail.com (E.L.F.); 2Institute of Biological and Health Science, Federal University of Mato Grosso, Barra do Garças 78600-000, MT, Brazil; daniellechfranca@gmail.com (D.C.H.F.); pgfmarchi.ufmt@gmail.com (P.G.F.d.M.); 3Postgraduate Program in Public Policies and Local Development, Escola Superior de Ciências da Santa Casa de Misericórdia de Vitória EMESCAM, Vitória 29045-402, ES, Brazil; tassiane.morais@emescam.br

**Keywords:** macrophages, colostrum, hormone, breast tumor

## Abstract

Human colostrum and milk contain diverse cells and soluble components that have the potential to act against tumors. In breast cancer, macrophages play a significant role in immune infiltration and contribute to the progression and spread of tumors. However, studies suggest that these cells can be reprogrammed to act as an antitumor immune response. This study aimed to evaluate the levels of melatonin and its receptors, MT_1_ (melatonin receptor 1) and MT_2_ (melatonin receptor 2), in colostrum and assess the differentiation and polarization of the colostrum macrophages modulated by melatonin in the presence of breast tumor cells. Colostrum samples were collected from 116 mothers and tested for their melatonin and receptor levels. The colostrum cells were treated with or without melatonin and then cultured for 24 h in the presence or absence of breast tumor cells. The results showed that melatonin treatment increased the expression of MT_1_ and MT_2_ in the colostrum cells. Furthermore, melatonin treatment increased the percentage of M1 macrophages and decreased the percentage of M2 macrophages. When the colostrum macrophages were cocultured with breast tumor cells, melatonin reduced the percentage of both macrophage phenotypes and the cytokines tumor necrosis factor-alpha (TNF-α) and interleukin 8 (IL-8). These data suggest that melatonin can regulate the inflammatory process via M1 macrophages in the tumor microenvironment and, simultaneously, the progression of M2 macrophages that favor tumorigenesis.

## 1. Introduction

Breast tumors have a complex microenvironment that includes resident tissue cells, such as adipocytes, and recruited cells, particularly immune system cells [1]. Macrophages are a major component of the immune infiltrate in breast cancer. While they can contribute to the progression and spread of tumors, studies suggest that these cells can also be reprogrammed to act as a potent antitumor immune response [2].

Macrophages can be polarized into two phenotypes, M1-like macrophages and M2-like macrophages, in response to signals from the microenvironment. These dynamic cells can participate in inflammatory processes or resolve diseases [3]. M1 and M2 macrophages express distinct surface markers and inflammatory mediators [4], and the phenotypic diversity of these cells increases with tumor development [5]. In tumors, the switch from the pro-immune M1-like macrophages to the immunosuppressive M2-like macrophages increases tumorigenesis and tumor progression and growth [6] via the release of cytokines [4,7].

Bioactive components are present in human colostrum and milk, such as antibodies, enzymes, cytokines, a large number of macrophages (55 to 60%), and the melatonin (MLT) hormone [8,9,10]. Melatonin has been found to stimulate the immune system and play a role in various physiological processes [11,12,13]. It acts through melatonin receptor 1A (MTRN1A) and melatonin receptor 1B (MTRN1B), also MT_1_ and MT_2_ receptors, respectively, in different organs, tissues, and cells, such as breast epithelial cells and immune cells [14,15,16].

Despite being present in all the body, there is still much to learn about the functional role of melatonin in humans. Studies have shown that melatonin can have an antitumor effect on human MCF-7 cells by increasing intracellular calcium release and inducing apoptosis [17]. Additionally, several intrinsic suppressor mechanisms can induce apoptosis to prevent the development of breast tumors [18,19].

Studies have shown that when monocytes or colostrum macrophages are cocultured with breast tumor cell lines, they can increase apoptosis and have antitumor effects [20,21,22]. In colostrum, macrophages comprise 40 to 50% of the cell content. However, the literature has not clarified the presence of the M1 and M2 phenotypes and their correlation with breast cancer. Colostrum macrophages differ morphologically and functionally from other monocytes and macrophages in other tissues [23,24].

This study explores the potential anti-cancer effects of human milk and melatonin. Many cells in human milk may have a pro-immune profile and could act on breast cancer when stimulated by melatonin. This work examines how melatonin affects the differentiation and polarization of colostrum macrophages in the presence of breast tumor cells (MCF-7).

## 2. Results

The melatonin levels were determined in colostrum supernatant, and the MT_1_ and MT_2_ concentrations were measured in colostrum cells lysed in the presence or not of melatonin (Table 1). MT_1_ and MT_2_, when treated with melatonin, showed higher concentrations in the lysed cells. Furthermore, the ratio of melatonin to MT_1_ in the lysed cells was higher than that of melatonin to MT_2_, as shown in Table 1.

Table 2 shows the percentage of colostrum macrophages CD14+ expressing CD163+ treated with melatonin. The results indicate the percentage of cells expressing CD14+ decreased after melatonin treatment. The expression of CD14+ in the colostrum cells was higher after 24 h of incubation. Moreover, the highest percentages of colostrum cells expressing CD14+ were observed in the cells not treated with melatonin after 24 h (Table 2).

Figure 1 shows the interactions of the colostrum macrophages and the MCF-7 cells in culture at 2 and 24 h in the presence or absence of the melatonin hormone. The most morphological alterations were observed after 24 h in culture.

In contrast, Table 2 shows that the expression of CD14 (a classic marker of monocytes/macrophages) and CD163^+^ (a phenotypic marker of M2 macrophages) in colostrum macrophages treated with melatonin decreased after 24 h in culture. On the other hand, the highest percentages of colostrum macrophage expression of CD14+CD163+ were observed in the cells not treated with melatonin.

Table 2 also displays the percentage of colostrum macrophages expressing CD163+ treated with melatonin in coculture with MCF-7 cells. It was observed that CD14+ expression decreased after melatonin treatment, although CD14+ expression in the colostrum cells in culture with MCF-7 cells was higher after 24 h of incubation regardless of the melatonin treatment (Table 2). Interestingly, the expression of CD14+CD163+ in the colostrum macrophages cocultured with MCF-7 cells was similar regardless of the incubation time or melatonin treatment (Table 2, Figure 2).

Table 3 compares the percentage of M1 and M2 expression in colostrum macrophages under different incubation conditions with and without melatonin. After 2 h in culture, the expression of M1 macrophages was lower in the colostrum cells, although treatment with melatonin increased their expression to levels similar to those of M2 macrophages. There was no difference in the expression of melatonin-treated M2 cells after 2 h of culture. However, after 24 h of culture, there was an increase in M1 macrophages and a decrease in M2 macrophages regardless of the melatonin treatment (Table 3 and Figure 3).

Additionally, Table 3 and Figure 3 display the expression of M1 and M2 cells treated with melatonin in culture with MCF-7 cells under different incubation conditions. After 24 h, melatonin reduced the expression of both M1 and M2 cells, although after 2 h, there was no difference in the expression of either cell type regardless of the melatonin treatment.

Table 4 shows the correlation between M1 and M2 cells in the presence or absence of melatonin. In addition, there was a positive correlation between M2 macrophages not treated with melatonin and M2 macrophages treated with the hormone.

Regarding the cytokine concentrations, it was observed that in the colostrum macrophages, melatonin increased the secretion of interleukin 8 (Table 5). On the other hand, the concentration of TNF-α and IL-8 in the culture supernatant of colostrum macrophages under melatonin modulation was reduced when cocultured with MCF-7 cells. For the other cytokines, no differences were observed (Table 5, Figure 4).

## 3. Discussion

Colostrum and human milk have been shown to reduce the risk of breast cancer [25] because they help to differentiate breast tissue and reduce the number of ovulatory cycles over a person’s lifetime. Reproductive factors can cause lasting changes in the epithelium of the mammary gland or surrounding stromal tissue, making the breast more or less susceptible to carcinogens [26,27].

Several studies have shown the interaction between immune system cells and tumor progression. In addition, the immune response to the tumor appears to be associated with the type of cell involved and hormonal interactions. In this study, CD163 expression and polarization of the M1 and M2 macrophages and cytokines released in culture supernatant in the presence of melatonin and breast tumor cells were evaluated in colostrum.

Studies have shown a relationship between the concentration of endogenous melatonin and the risk of breast cancer in humans. Specifically, research has suggested that melatonin [17,28] and its receptors have oncostatic properties in breast cancer models [29,30]. In this study, the melatonin levels showed concentrations similar to other works in the literature [12,31].

The effect of melatonin on the body depends on several factors, including the concentration, circadian rhythm, and receptor affinity. It has been shown that melatonin and its receptors can impact cancer on a cellular level. Specifically, the MT_1_ melatonin receptor has been found to play a role in the anti-cancer effect of melatonin in breast cancer models. Interestingly, this receptor may be continuously active in breast cancer cells, naturally suppressing their growth [16]. The administration of exogenous melatonin can cause changes in the expression and function of MT_1_ and MT_2_ receptors. This has been observed in mammals, with research indicating that the amount of melatonin can impact receptor expression and function. A study using a system of melatonin receptors found that the dosage of melatonin influenced its expression and function [32]. This study detected both melatonin receptors (MT_1_ and MT_2_) in colostrum cell lysate, with their concentrations being higher in the presence of exogenous melatonin with a higher melatonin and MT_1_ receptor ratio, indicating that melatonin interacts with colostrum cells, particularly mononuclear cells, which probably can regulate macrophage functions and the antitumor effects.

Macrophages are cells that play a role in both pro- and anti-inflammatory responses in the microenvironment of malignant tumors [33]. Depending on their activation state, they can transform into different specific phenotypes [34]. Different factors can impact the phenotype of macrophages, which in turn affects their performance. Activated macrophages fall into M1-like and M2-like macrophages [35].

M1 macrophages promote inflammation and have strong microbicidal activity [36,37], while M2 macrophages have anti-inflammatory properties and produce cytokines that neutralize inflammation [38]. This study showed CD163-positive macrophages in colostrum, and the hormone melatonin reduced the percentage of these cells after 24 h in culture. CD163 is a phenotypic marker of M2 macrophages and can be used to distinguish M2 and M1 macrophages.

Macrophages express the marker CD163+ during the resolving phase of inflammation [34,39]. Studies have shown that the proportion of these cells increases after 40 h in culture, indicating a resolving macrophage phenotype [40]. However, when colostrum macrophages were cocultured with MCF-7 cells and treated with melatonin, the expression of this marker remained similar regardless of the incubation time. Despite not causing inflammatory processes, colostrum macrophages are still considered activated cells, which have high phagocytic and microbicidal activity when stimulated by melatonin [12,28,41] and can produce oxygen-free radicals [23,42]. Interestingly, human colostrum supernatant can change murine macrophages to the M2 profile in vitro, as shown in a study by Panahipour et al. [43], reinforcing the hypothesis that the components found in human milk have anti-inflammatory properties.

In this study, the expression of M1 macrophages was lower in colostrum, and melatonin restored the expression of M1 cells to percentages similar to those of M2. However, when in culture for 24 h, independent of melatonin, an increase in M1 macrophages and a reduction in M2 macrophages were observed. Interestingly, there was a positive correlation between M2 macrophages not treated with melatonin and M2 macrophages treated with the hormone, suggesting the action of this hormone in the M2 immunosuppressive phenotype.

Due to macrophage plasticity, studies have revealed that melatonin modulates the polarization of M1 macrophages via the STAT-1, NF-kB, and NLRP3 pathways and of M2 macrophages via STAT6 activation [44,45], being able to modulate the development and progression of several diseases associated with macrophages [40], including breast cancer [28]. Experimental and systems biology modeling studies have revealed that the polarization spectrum of M1–M2 macrophages results from a complex network of conduction signaling pathways and intracellular regulatory mechanisms. This allows for investigating system-level mechanisms during macrophage polarization and repolarization. [46].

In breast cancer, macrophages represent one of the main components of the immune infiltrate, and these cells appear to contribute to the progression and spread of tumors. Still, studies suggest that these cells can be reprogrammed to act as a potent antitumor immune response [2]. Macrophages are dynamic and participate in inflammatory processes or disease resolution [3], and the phenotypic diversity of these cells increases with tumor development [5].

In tumors, the switch from the pro-immune M1 phenotype to the immunosuppressive M2 phenotype increases tumorigenesis and tumor progression and growth [6] by releasing cytokines and growth factors [4,47,48]. Here, melatonin reduced the expression of both macrophage phenotypes (M1 and M2) in culture with MCF-7 cells after 24 h, suggesting that this hormone acts on both macrophage phenotypes. This effect is important for the antitumor action of this hormone [17] since it reduces both the pro-immune M1 and the immunosuppressive M2 phenotype in the breast microenvironment.

Various cytokines can be released into the tumor environment against or favoring tumor growth. For instance, cytokines like IL-1β, IL-6, IL-8, and TNF-α play a crucial role in breast cancer. Several studies indicate that these cytokines can increase the transcriptional levels of various inflammatory factors and chemokines, potentially leading to tumor formation [49,50,51,52]. However, oncogenic and tumor suppressor effects have also been observed [53]. IL-6 has physiological and pathological roles in inflammation and immunity and can promote tumor growth. A deregulated IL-6 signaling pathway has been found to play an important role in tumor proliferation, migration, and adhesion [54]. Moreover, in an inflammatory tumor microenvironment, IL-1β stimulates IL-6 production, which can increase the aggressiveness of luminal-type breast cancer cells [55].

The anti-inflammatory cytokine IL-10 can also unexpectedly stimulate tumor cell proliferation and migration [56]. According to this study, melatonin caused a decrease in the amount of TNF-α found in the culture supernatant of colostrum macrophages and MCF-7 cells. M1 macrophages release TNF-α and can promote tumor formation by increasing the inflammatory factor and chemokine transcriptional levels [57,58]. Other studies have shown that melatonin has antitumor and anti-inflammatory effects [58,59,60,61]. The results of this study support these findings, as the secretion of TNF-α and expression of M1 macrophages were reduced in cocultures treated with melatonin; this hormone could be a potential therapeutic strategy for cancer.

Interestingly, the melatonin-treated cocultures also showed a decrease in IL-8 secretion. Interleukin 8 is a cytokine that is also pro-inflammatory [59,60], whose expression is closely related to the development of several tumors, in particular breast cancer [61,62,63]. In addition, IL-8 is one of the components of colostrum and mature milk, showing a higher concentration in the early stages of lactation [64,65].

Cytokines, such as IL-8, are crucial for regulating the immune response [66]. IL-8 is important during the early phase of inflammation, as it activates and attracts neutrophils [34,67]. However, IL-8 is a significant factor in regulating the microenvironment in the context of a tumor microenvironment [7,63,68]. Kim et al. [69] showed that melatonin suppressed IL-8 production in human lung fibroblasts, reinforcing the antitumor role of melatonin. Further studies are required to comprehend the actual mechanisms involved in this process.

## 4. Materials and Methods

A cross-sectional study was carried out with the participation of 116 clinically healthy donors enrolled at the Municipal Hospital of Barra do Garças, MT, Brazil. The study’s inclusion criteria were as follows: aged 18 to 35 years; gestational age at delivery between 37 and 41^6/7^ weeks; negative serological reactions for hepatitis, HIV, and syphilis during prenatal period; no food restrictions; and signed informed consent form. The exclusion criteria were: gestational diabetes; type 1 and type 2 diabetes mellitus; obesity; twin pregnancy; fetal malformations; and delivery before the 36th week of gestation.

### 4.1. Colostrum Sampling and Separation of Colostral Cells

Approximately 8 mL of colostrum from each woman was collected in sterile plastic tubes between 48 and 72 h postpartum. The samples were centrifuged (160× *g*, 4 °C) for 10 min, separating the colostrum into three phases: a cell pellet, an intermediate aqueous phase, and a lipid-containing supernatant. The upper fat layer was discarded, and the aqueous supernatant was stored at −80 °C for later analyses. The cells were separated via a Ficoll–Paque gradient (Pharmacia, Upsala, Sweden), producing 98% pure mononuclear cells prepared and analyzed via light microscopy. The purified macrophages were resuspended independently in serum-free medium 199 at a 2 × 10^6^ cells/mL final concentration. The cells were used for assays of immunophenotyping and MT_1_ and MT_2_ receptors. The scheme for obtaining samples and the experimental design is shown in Figure 5.

### 4.2. Melatonin Determination

The melatonin hormone in colostrum supernatant was quantified via enzyme immunoassay on ELISA microplates. The test had a lower detection limit of 1.6 pg/mL, and the coefficients of variation (%) for the intra-assay and inter-assay were 3.0–11.4 and 6.4–19.3, respectively. The colostrum melatonin was extracted via affinity chromatography and concentrated in a vacuum centrifuge. The ELISA kit we used was an Immuno-Biological Laboratories Melatonin ELISA RE54021-IBL, Hamburg, German. The reaction values were measured using a plate spectrophotometer with a 405 nm filter and the results were obtained through a standard curve, which we expressed in pg/mL.

### 4.3. Melatonin Receptor 1A (MTNR1A) and Melatonin Receptor 1B (MTNR1B) Determination

Following the separation, some colostrum cells were treated or not with 50 µL of melatonin (MLT) at a final concentration of 100 nM [12]. The cells were then incubated for 24 h at 37 °C with 5% CO_2_. After incubation, the cells were resuspended in RPMI 1640 medium containing 10% fetal bovine serum (FBS-Sigma, St. Louis, MO, USA) and lysed with Triton X 100 (Sigma, St. Louis, MO, USA) for 5 min. The lysed cells were centrifuged at 500× *g* for 10 min at 8 °C to remove cellular debris. A sandwich enzyme immunoassay (ELISA kit—Cloud Clone Corporation, Katy, TX, USA) for the in vitro quantitative measurement of MT_1_ (MTNR1A) and MT_2_ (MTNR1B) receptors in human colostrum cells lysed was used and the procedure was performed according to the manufacturer’s instructions. For each assay performed, a phagocyte control (2 × 106 cells/mL) was incubated for a similar time, depending on the type of assay in the medium 199 in the absence of melatonin. The ELISA kit for both MTNR1A (MT_1_) and MTNR1B (MT_2_) was used to present the following characteristics: the lower detection limit was 156 pg/mL (0.156 ng/mL) and the intra-assay and inter-assay coefficients of variation (%) were <10% and <12%, respectively. The reaction values were measured via absorbance in a plate spectrophotometer with a 450 nm filter. The results were obtained through a standard curve and expressed in pg/mL.

### 4.4. Tumor Cell Lines and Cell Culture

ATCC (American Type Culture Collection, Manassas, VA, USA) breast adenocarcinoma (MCF-7) cell lines were used to analyze the differentiation and polarization of colostrum macrophages modulated by melatonin in the presence of breast tumor cells.

The lines were cultivated and frozen in liquid nitrogen for storage. Subsequently, they were cultivated in a 1:1 mixture of DMEM and Ham’s F10 (Sigma), plus HEPES, penicillin, streptomycin, sodium bicarbonate, and fetal bovine serum (SFB, Cultilab, Campinas, SP, Brazil). The cells were grown in bottles and kept at 5% CO_2_ and a temperature of 37 °C until the formation of a cell monolayer. Then, the cell bottles were subjected to trypsinization. First, the cells were washed with 5 mL Hanks, and the strains were subjected to 3 mL of trypsin 0.05% EDTA produced from a 10-fold concentrated solution (Invitrogen, Waltham, MA, USA) until the cells were detached from the bottles. Next, the cells were homogenized with varying volumes of the culture medium plus 10% fetal bovine serum for trypsin neutralization. Finally, the volume of the cell suspension obtained in one bottle was transferred to two other bottles to obtain an adequate cell quantity for the experiments. Next, the concentrations of MCF-7 cells were adjusted to 1 × 10^5^ cells/mL, and they were subcultured in RPMI medium plus 10% fetal bovine serum in 24-well plates in the presence or absence of 50 μL of melatonin for 24 h at 5% CO_2_ and a temperature of 37 °C. A control was performed using MCF-7 cells in the absence of treatment. The MCF-7 cells were also cocultured with colostrum cells (2 × 10^6^ cells/mL) and subcultured using the same protocol described above.

After this period, the MCF-7 cells cocultured or not with colostrum cells were trypsinized again, washed twice, and submitted to different assays. In addition, the coculture cells were centrifuged and pellet dyed with 200 µL acridine orange (Sigma, St. Louis, MO, USA; 14.4 g/L) for 1 min. The sediment was resuspended in cold 199 medium, washed twice, and observed under immunofluorescence microscopy at 400× and 1000× magnification.

### 4.5. Melatonin Hormone Modulation

The colostrum cells were incubated with melatonin in the presence or absence of MCF-7 cells and kept in culture for 2 and 24 h. For each assay performed, as a control for the experiments, the cells (2 × 10^6^ cells/mL) were incubated for a similar time, depending on the type of assay, in medium 199 in the absence of the hormone. The final melatonin concentration was 100 nM, as previously determined [12].

### 4.6. Immunophenotyping and Macrophage Identification and Polarization

The colostrum cells were washed with Phosphate Buffer (PBS) plus BSA (bovine serum albumin) for 10 min at 4 °C. The cells were labeled with 5 μL of anti-CD14+(FITC). A PE-tagged IgG1 isotype was used as a control. The cells were evaluated via flow cytometry. The cells expressing CD14+ were used for the polarization analysis [33,61,62]. The cell suspensions were labeled with antibodies specific for CD197, CD86, and CD163, and they were fixed and permeabilized with Cytofix–Cytoperm solution (BD, Bioscience, San Jose, CA, USA). The cells that were CD197+CD86+ were defined as M1-type macrophages, while the cells that were CD197-CD86+ or CD14+CD163+ were defined as M2-type macrophages [70,71].

### 4.7. Cytokine Determination

The cytokine concentrations in the culture of supernatant were evaluated using the Cytometric Bead Array Kit (CBA, BD Bioscience, San Jose, CA, USA). The IL-1-β, IL-6, IL-8, IL-10, TNF-α, and IL-12p70 concentrations were analyzed according to the manufacturer’s instructions. These cytokines were analyzed using flow cytometry (FACSCalibur, BD Bioscience, San Jose, CA, USA). The results were generated in graphs and tables using the CellQuest (BD)^®^ software version 5.1, and the data were analyzed using the FCAP Array software version 3.0.

### 4.8. Statistical Analysis

Data are expressed as the mean ± standard deviation. Statistically significant differences in the melatonin receptors and percentages of CD197+, CD86+, and CD163+ cells treated or not treated with melatonin and cocultured or not with MCF-7 cells were evaluated using an analysis of variance (ANOVA), and the correlations between the M1 and M2 macrophages were evaluated using Pearson’s linear correlation. Statistical significance was considered for a *p*-value < 0.05.

## 5. Conclusions

These data suggest that colostrum macrophages can differentiate into M1 or M2 phenotypes and that the hormone melatonin increases the number of M1 macrophages known to protect against pathogens. When these macrophages are cocultured with MCF-7 cells, melatonin reduces the number of M1 and M2 macrophages, suggesting that it may prevent the formation of the inflammation that M1 macrophages cause in the tumor microenvironment and, at the same time, the progression of M2 macrophages, which promote tumorigenesis. Furthermore, melatonin reduces the secretion of the pro-inflammatory cytokines TNF-α and IL-8 when cocultured with MCF-7 cells, which are closely related to tumor development and reinforce the potential of melatonin as a therapeutic strategy for cancer.

## Figures and Tables

**Figure 1 ijms-24-12400-f001:**
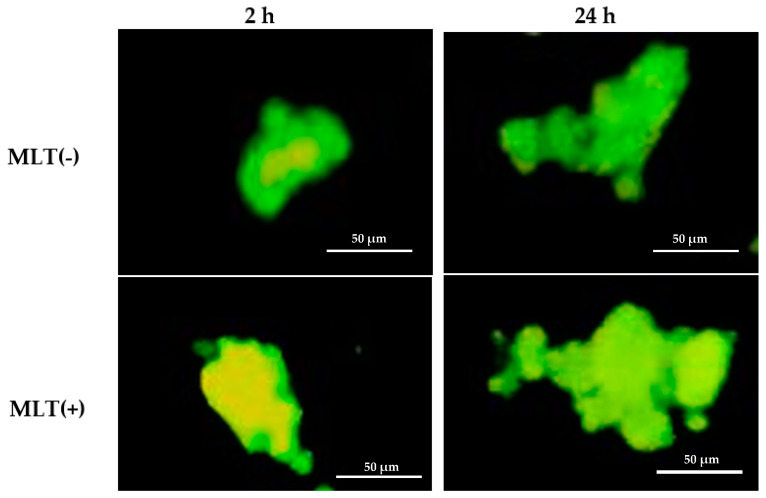
Illustrative images of the coculture of colostrum macrophages and MCF-7stained via acridine orange methods and analyzed by fluorescence microscopy. The colostrum cells and MCF-7 cells were incubated with melatonin for 2 and 24 h. Without melatonin MLT (−); with melatonin MLT (+).

**Figure 2 ijms-24-12400-f002:**
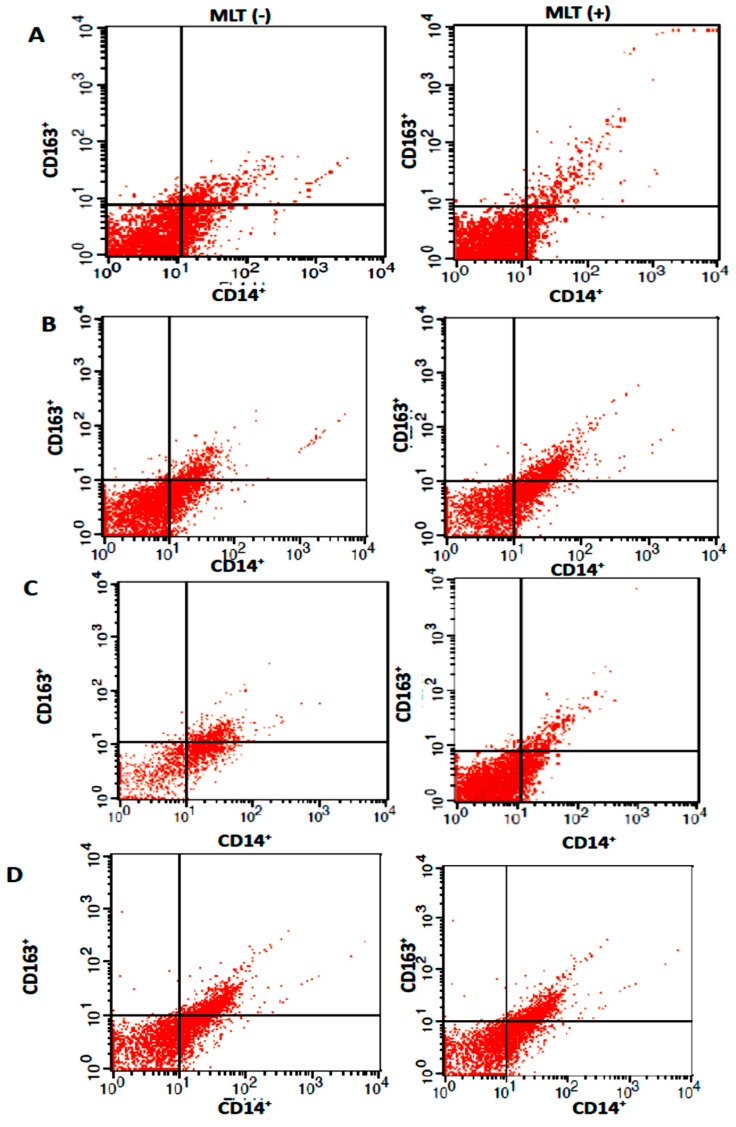
Flow cytometric analysis of subsets of cells expressing CD14^+^ (colostrum macrophages) and CD14^+^CD163+ (M2 macrophages) at 2 ((**A**) colostrum cells and (**B**) coculture figures) and 24 h ((**C**) colostrum cells and (**D**) cocultures figures) in the presence (MLT +) or not of melatonin (MLT −).

**Figure 3 ijms-24-12400-f003:**
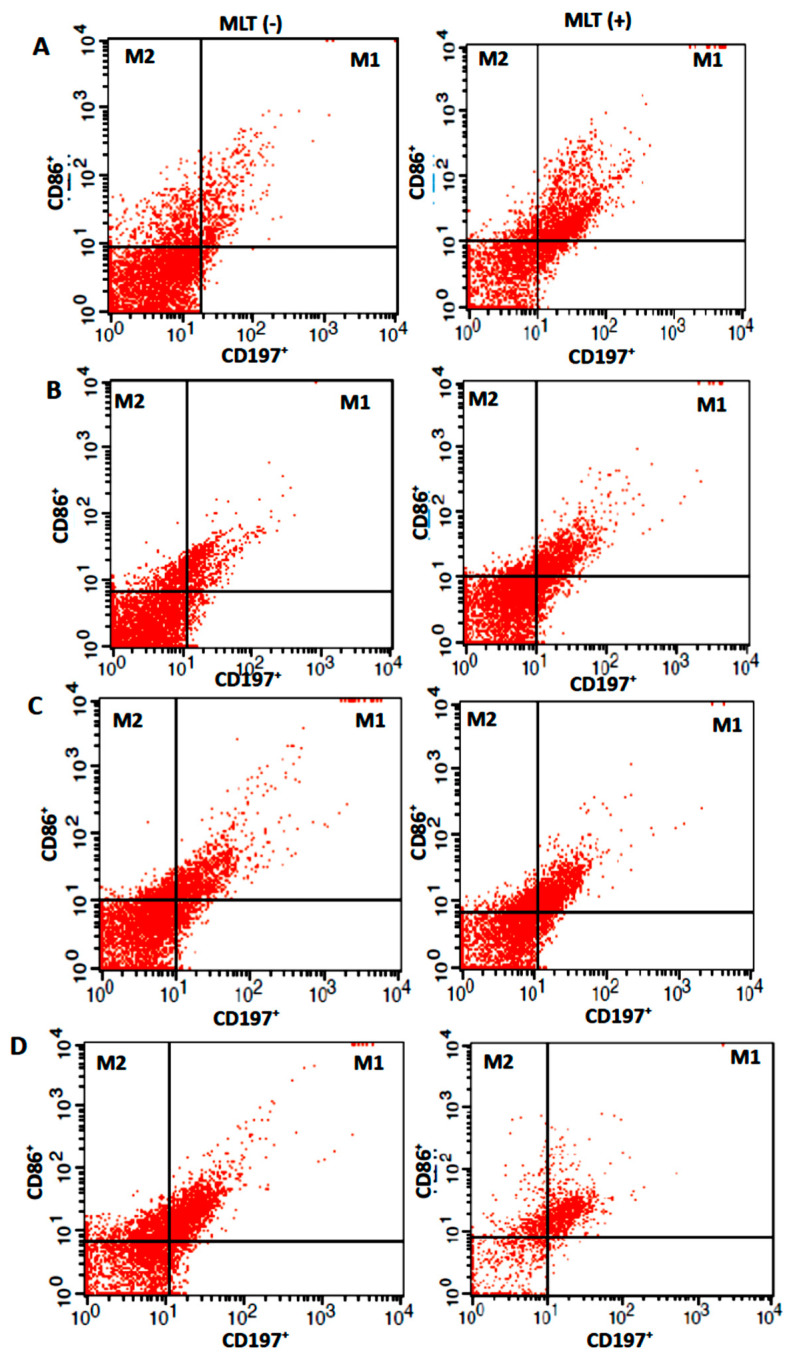
M1 and M2 expression in (**A**) colostrum cells non-treated, (**B**) treated with melatonin (MLT), and (**C**) colostrum cells in coculture with MCF-7 non-treated and (**D**) treated with melatonin (MLT) indicated by fluorescence intensity. Flow cytometry (FACSCalibur, Becton Dickinson, USA) carried out immunofluorescence assays. Without melatonin MLT (−); with melatonin MLT (+).

**Figure 4 ijms-24-12400-f004:**
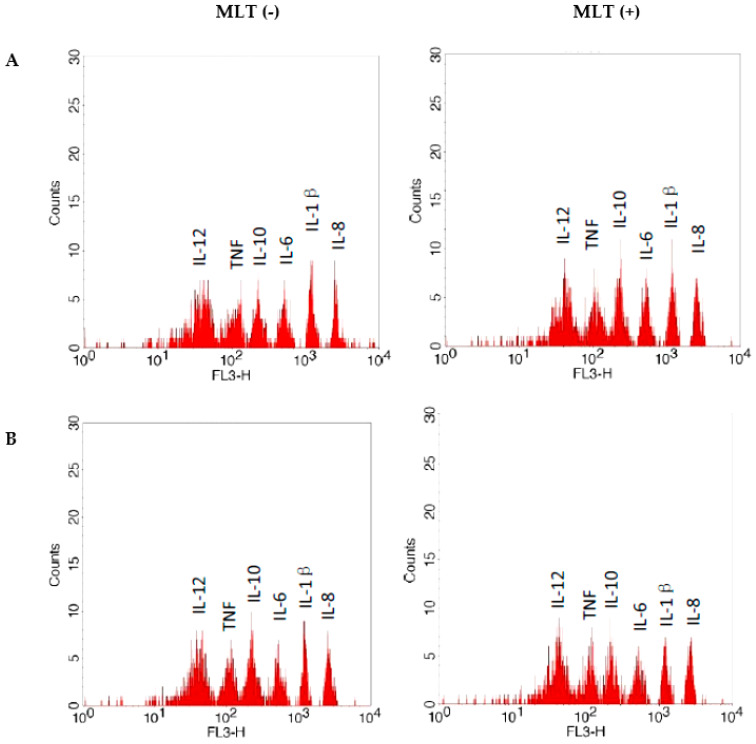
Fluorescence intensity of cytokines in culture supernatants (**A**) colostrum cells and (**B**) coculture colostrum cells and MCF-7 treated or not with melatonin (MLT). Fluorescence analyses were carried out via flow cytometry (FACSCalibur, Becton Dickinson, USA). FL3 (fluorescence in channel 3). Without melatonin MLT (−); with melatonin MLT (+).

**Figure 5 ijms-24-12400-f005:**
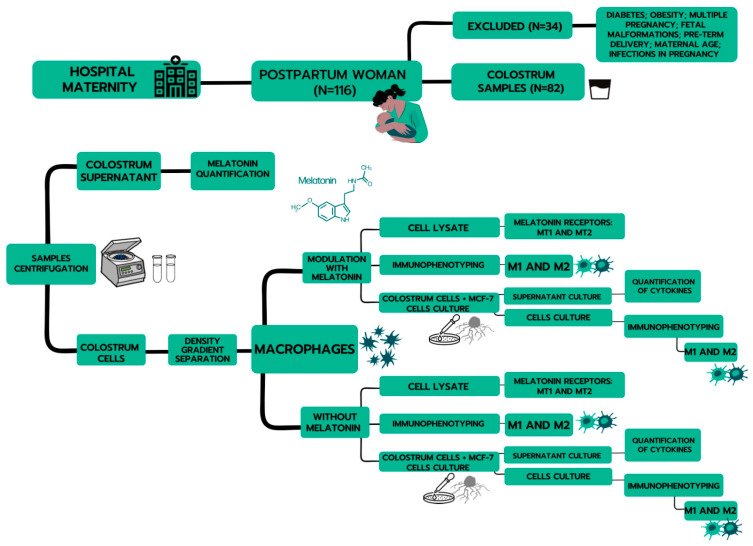
Representative scheme for obtaining samples and the experimental design.

**Table 1 ijms-24-12400-t001:** Melatonin concentrations in colostrum supernatant and MT_1_ and MT_2_ concentrations in colostrum cells lysed treated or not with melatonin.

Parameters	pg/mL
MLT	183 ± 33
MT_1_ in cells lysed 199 medium	277 ± 135
MT_1_ cells lysed + MLT	850 ± 138 *
MT_2_ in cells lysed medium 199	670 ± 58
MT_2_ in cells lysed + MLT	952 ± 110 *
MLT/MT_1_ cells lysed ratio	0.7 ± 0.2
MLT/MT_2_ cells lysed ratio	0.3 ± 0.1 #

Note: MLT-melatonin; MT_1_-melatonin receptor 1; MT_2_-melatonin receptor 2. *p* < 0.05. * differences between melatonin receptor (MT_1_ or MT_2_) treated or not with melatonin; # differences between MLT/receptor (MT_1_ or MT_2_) in cells lysed.

**Table 2 ijms-24-12400-t002:** Expression percentage of the macrophage scavenger receptor CD163+ in colostrum macrophages in coculture or not with MCF-7 cells and treated or not with melatonin.

	Colostrum	Melatonin	2 h	24 h
	CD14+	No	10.8 ± 4.7	26.9 ± 7.0 +
Cells		Yes	6.5 ± 1.9 *	16.4 ± 7.1 *+
	CD14+CD163+	No	15.0 ± 2.5	26.3 ± 6.1 +
		Yes	16.3 ± 7.0	16.9 ± 7.1 *
	CD14+	No	10.7 ± 5.7	22.5 ± 5.2 +
Coculture		Yes	6.4 ± 2.0 *	23.7 ± 5.9 +
	CD14+CD163+	No	15.0 ± 2.5	16.1 ± 6.6
		Yes	16.2 ± 7.7	15.6 ± 6.4

Note: *p* < 0.05. * differences between untreated and melatonin-treated cells, considering the same incubation time; + differences between incubation times, considering the same treatment.

**Table 3 ijms-24-12400-t003:** Expression in percentage (%) of M1 and M2 in colostrum cells in coculture or not with MCF-7 cells in the presence of melatonin at different incubation times.

	Colostrum	Melatonin	2 h	24 h
	M1	No	11.3 ± 5.5	31.0 ± 2.2 +
Cells		Yes	21.4 ± 7.1 *	27 ± 4.0
	M2	No	28.5 ± 4.2 #	4.8 ± 1.4 #+
		Yes	22.3 ± 7.2	5.5 ± 1.8 #+
	M1	No	24.4 ± 5.2	24.1 ± 9.4
Coculture		Yes	18.3 ± 5.0	3.5 ± 0.5 *+
	M2	No	24.8 ± 10.0	24.8 ± 10.0
		Yes	17.1 ± 4.3	5.2 ± 2.0 *+

Note: *p* < 0.05. * differences between untreated and melatonin-treated cells, considering the same cell type and incubation time; # differences between cell types, considering the same treatment and incubation time; + differences between incubation time, considering the same treatment and cell type.

**Table 4 ijms-24-12400-t004:** Correlation between M1 and M2 cells in the presence or absence of melatonin.

Macrophages	M1 MLT (−)	M1 MLT (+)	M2 MLT (−)	M2 MLT (+)
**M1 MLT (−)**	__	r = 0.4709 *p* = 0.3458	r = −0.2864 *p* = 0.9821	__
**M1 MLT (+)**	r = 0.4709 *p* = 0.3458	__	__	r = 0.7231 *p* = 0.1043
**M2 MLT (−)**	r = 0.2864 *p* = 0.9821	__	__	r = 0.7098 *p* = 0.0498
**M2 MLT (+)**	__	r = 0.7231 *p* = 0.1043	r = −0.7098 *p* = 0.0498	__

r = Pearson’s correlation coefficient; *p* (*p*-value).

**Table 5 ijms-24-12400-t005:** Cytokine concentrations in the culture supernatant of colostrum macrophages in coculture or not with MCF-7 untreated or treated with melatonin.

Cells	MLT	Time	IL-1 β	IL-6	IL-8	IL-10	TNF-α	IL-12P70
**MN**	No	24 h	55.5 ± 17.2	10.7 ± 10.4	85.1 ± 68.5	28.4 ± 6.7	13.5 ± 7.5	23.3 ± 15.
	Yes	24 h	43.3 ± 19.8	14.1 ± 9.3	102.5 ± 78.0 *	31.3 ± 7.3	22.2 ± 7.4 #	39.2 ± 8.6
**MN+MCF-7**	No	24 h	48.2 ± 17.5	16.9 ± 8.2	64.7 ± 12.1 #	33.8 ± 9.6	15.7 ± 5.9	28.7 ± 15.5
	Yes	24 h	47.4 ± 12.6	17.5 ± 8.9	41.5 ± 5.7 #*	27.7 ± 2.6	10.8 ± 6.4 #	22.1 ± 16.3

Note: MN-mononuclear cells; IL-interleukin. *p* < 0.05. * Differences between colostrum cells plus MCF-7 not treated and treated with melatonin, considering the same incubation time; # differences between colostrum cells and coculture with MCF-7, both treated with melatonin and in the same incubation time.

## Data Availability

The authors will make the data supporting this study’s interpretations available if requested.

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
