# Peer review of "Polarization of Melatonin-Modulated Colostrum Macrophages in the Presence of Breast Tumor Cell Lines"

_ijms, 2023, doi:10.3390/ijms241512400_

Round 1
Reviewer 1 Report
The authors presented an investigation that studies macrophage polarization under melatonin treatment, which may potentially be influenced if co-cultured together with breast cancer cells. There are a couple of points for the authors to consider to improve this manuscript.
1. In Table 1, why melatonin receptors show higher levels in lysed cells when treated with melatonin?
2. In Table 1, expression of melatonin receptors in lysed cells by ELISA does not necessarily indicate expression on cell surfaces, which are needed for melatonin signaling. Flow cytometry is more appropriate.
3. In Tables 2&3, what is the unit of expression? Percentage or something else?
4. What is the physiological meaning of a macrophage being CD14+ vs CD14+CD163+? The authors should describe this in lines 91-94.
5. In Table 3, description of macrophage phenotypes were changed to M1 vs M2. The authors should explain what are the markers used to characterize M1 vs M2.
6. What is r and P in table 4? The authors should explicitly describe the meaning of correlation in Table 4.
7. The relationship between colostrum and cancer is weak. Why is colostrum linked to cancer? What is the physiological/pathophysiological meaning behind it?
8. Macrophage polarization is now recognized as no longer a dichotomous process; instead researchers think of macrophages being more M1-like and M2-like which spans a spectrum. This has been illustrated by a number of experimental and systems biology modeling studies (e.g. PMID 33659877), which should be discussed.
writing should be improved.
Author Response
reviewer 1.
Thank you very much for the suggestions for improving the manuscript. All changes have been made and are described below.
Comments and Suggestions for Authors
The authors presented an investigation that studies macrophage polarization under melatonin treatment, which may be influenced if co-cultured with breast cancer cells. There are a couple of points for the authors to consider to improve this manuscript.
- In Table 1, why do melatonin receptors show higher levels in lysed cells when treated with melatonin?
R: The administration of exogenous melatonin can cause changes in the expression and function of MT1 and MT2 receptors. Exposing these receptors to melatonin can affect their expression in mammals. A study using a heterologous expression system of melatonin receptors found that the expression and function changed depending on the dose of exogenous melatonin. In this study, the increased concentration of both receptors was probably due to the action of exogenous melatonin at a dose of 100nM.
This information was included in the discussion section.
- Gether, J.A. Ballesteros, R. Seifert, E. Sanders-Bush, H. Weinstein, B.K. Kobilka
Structural instability of a constitutively active G protein-coupled receptor. Agonist-independent activation due to conformational flexibility
- Biol. Chem., 272 (1997), pp. 2587–2590.
Gerdin MJ, Masana MI, Dubocovich ML. Melatonin-mediated regulation of human MT(1) melatonin receptors expressed in mammalian cells. Biochem Pharmacol. 2004 Jun 1;67(11):2023-30. doi: 10.1016/j.bcp.2004.01.027. PMID: 15135299.
- In Table 1, expression of melatonin receptors in lysed cells by ELISA does not necessarily indicate expression on cell surfaces, which are needed for melatonin signaling. Flow cytometry is more appropriate.
This study established the relationship between melatonin concentrations and receptors, so we evaluated receptor concentrations in lysed cells.
- In Tables 2&3, what is the unit of expression? Percentage or something else?
R: The cells were expressed in percentage in both tables and added to the text.
- What is the physiological meaning of a macrophage being CD14+ vs CD14+CD163+? The authors should describe this in lines 91-94.
CD14+ is a cellular co-receptor for lipopolysaccharide (LPS) and is considered a classic marker for monocytes and macrophages, and CD163+ is confirmed as a phenotypic marker of M2 macrophages and used to distinguish M2 and M1 macrophages during the resolution phase of an inflammatory process, tissue injury or even in a tumor microenvironment because such environments are rich in macrophages with this phenotype.
This information was added according to suggest now in lines 111-112.
Quero L.; Hanser E.; Manigold T.; Tiaden A. N.; Kyburz D. TLR2 stimulation impairs the anti-inflammatory activity of M2-like macrophages, generating a chimeric M1/M2 phenotype. Arthritis Res Ther. 2017, 2; 245. DOI: 10.1186/s13075-017-1447-1.
Landmann R.; Müller B.; Zimmerli W. CD14, new ligand, and signal diversity aspects. Microbes Infect. 2000, 2, 3, 295–304. DOI: 10.1016/s1286-4579(00)00298–7.
Souza, S.; Brion, R.; Lintunen, M.; Krongyist, P.; Sandholm, J.; Mönkkönen, J.; Kellokumpu-Lehtinen, P.; Lauttia, S.; Tynninen, O.; Joensuu, H.; Heymann, D.; Määttä, J. Human breast cancer cells educate macrophages toward the M2 activation status. Breast Can Res. 2015, 17, 101, 1–14. DOI: 10.1186/s13058-015-0621-0.
- In Table 3, the description of macrophage phenotypes was changed to M1 vs M2. The authors should explain what are the markers used to characterize M1 vs M2.
R: The markers used were: Macrophages expressing CD197+CD86+ were defined with M1-type macrophages, while CD197-CD86+ were defined as M2-type macrophages (Benoit; Desnues; Mege, 2008; Kwieciem et al., 2019; Jayasingam et al., 2020). This information is described in Materials and Methods
Jayasingam, S. D.; Citartan, M.; Thang, T. H.; Mat Zin, A. A.; Ang, K. C.; Ch’ng, E. S. Evaluating the Polarization of Tumor-Associated Macrophages Into M1 and M2 Phenotypes in Human Cancer Tissue: Technicalities and Challenges in Routine Clinical Practice. Front Oncol. 2020, 9, 1–9. DOI: 10.3389/fonc.2019.01512.
Benoit, M.; Desnues, B.; Mege, J. L. Macrophage Polarization in Bacterial Infections. J Immunol. 2008, 181, 3733–3739. DOI: 10.4049/jimmunol.181.6.3733.
Kwiecień I, Polubiec-Kownacka M, Dziedzic D, Wołosz D, Rzepecki P, Domagała-Kulawik J. CD163 and CCR7 as markers for macrophage polarization in lung cancer microenvironment. Cent Eur J Immunol. 2019; 44, 395–402. DOI: 10.5114/ceji.2019.92795.
- 6. What are r and P in Table 4? The authors should explicitly describe the meaning of correlation in Table 4.
R: Pearson's correlation coefficient (r) measures the degree of correlation between two metric scale variables; in this study, we used statistical analysis to verify the correlation between M1 and M2 cells in the presence or absence of melatonin. P is the P-value of Statistical significance. This information is described in materials and methods - 4.8 Statistical analysis.
- The relationship between colostrum and cancer is weak. Why is colostrum linked to cancer? What is the physiological/pathophysiological meaning behind it?
Although the mechanisms are not entirely elucidated, colostrum and human milk has been hypothesized to reduce the risk of breast cancer primarily through the differentiation of breast tissue and the reduction of the lifetime number of ovulatory cycles (Yang & Jacobsen, 2008). Reproductive factors may induce permanent changes in the mammary gland epithelium or surrounding stromal tissue (Russo et al., 2005; Russo et al., 2008). It is most likely that the tissue changes can make the breast more or less susceptible to carcinogenic factors (Russo et al., 2005).
Also, the colostrum includes components such as lysozyme, lactoferrin, peroxidase, complex oligosaccharides (receptor analogs), fatty acids (lipids), and mucins (Brandtzaeg, 2003; 2010), and a variety of leukocytes specially Macrophages (55- 60%) in elevated concentrations (Islam, 2006; Brandtzaeg, 2010). These important immunoregulatory factors can lead to future breast câncer protection.
Brandtzaeg P (2010). The mucosal immune system and its integration with the mammary glands. J Pediatr, 156, 8-15.
Islam N, Ahmed L, Khan NI, et al (2006). Immune components (IgA et al. immune cells) of colostrum of Bangladeshi mothers. Pediatr Int, 48, 543-8.
Russo J, Balogh GA, Russo IH (2008). Full-term pregnancy induces a specific genomic signature in the human breast. Cancer Epidemiol Biomarkers Prev, 17, 17-51.
Russo J, Moral R, Balogh GA, et al (2005). The protective role of pregnancy in breast cancer. Breast Cancer Res, 7, 131-41.
Yang L, Jacobsen KH (2008). A systematic review of the association between breastfeeding and breast cancer. J Womens Health, 17, 1635-45.
- Macrophage polarization is now recognized as no longer a dichotomous process; instead, researchers think of macrophages as being more M1-like and M2-like, which spans a spectrum. This has been illustrated by many experimental and systems biology modeling studies (e.g., PMID 33659877), which should be discussed.
R: This approach based on simulation studies and biological systems was added in the discussion of the paper comments on the Quality of English Language
Writing should be improved.
R: Thank you for the suggestions, and we apologize for any inconvenience. The text was improved and revised by a native English speaker.

Reviewer 2 Report
The study evaluated levels of melatonin and its receptors MT1 and MT2 in human colostrum and assessed the differentiation and polarization of colostrum macrophages modulated by melatonin in the presence of breast tumor cells. This study explores the potential anti-cancer effects of human milk and melatonin. The results are very interesting. There are several issues to address and amend:
1. All abbreviations in the Abstract should be deciphered.
2. It would be beneficial for the readers to see the images of co-cultured cells. Some macrophages can start to differentiate in the presence of cancer cells. Authors did not present those observations.
3. Images of MCF-7 cells at different time points could also be presented. It is unclear whether any morphological changes were observed.
4. Expression of MT receptors was not visualized using fluorescent microscopy or IHC. It is a significant fault of this study.
5. Flow cytometry data are not shown. Authors could present them.
6. Discussion: Authors wrote :“ … cytokines like IL-1β, IL-6, IL-8, and TNF-α play a crucial role.” It is necessary to clarify the role of different cytokines and cite relevant papers. For instance, TNFa kills cancer cells through death receptor signalling pathway. However, TNFa can promote cancer metastasis if death receptor pathway is not functional. The dual mechanisms should be accented.
Good English, minor editing is recommended.
Author Response
Reviewer 2
Thank you very much for the suggestions for improving the manuscript. All changes have been made and are described below.
Comments and Suggestions for Authors
The study evaluated melatonin levels and its receptors MT1 and MT2 in human colostrum. It assessed the differentiation and polarization of colostrum macrophages modulated by melatonin in the presence of breast tumor cells. This study explores the potential anti-cancer effects of human milk and melatonin. The results are very interesting. There are several issues to address and amend:
- All abbreviations in the Abstract should be deciphered.
The abbreviations are improved in the abstract.
- It would benefit the readers to see the images of co-cultured cells. Some macrophages can start to differentiate in the presence of cancer cells. The authors did not present those observations.
The figure was added in the result section.
- Images of MCF-7 cells at different time points could also be presented. It is unclear whether any morphological changes were observed.
The figure was added in the result section.
- Expression of MT receptors was not visualized using fluorescent microscopy or IHC. It is a significant fault of this study.
This study established the relationship between melatonin concentrations and receptors, so we evaluated receptor concentrations in lysed cells by ELISA.
- Flow cytometry data are not shown. Authors could present them.
The Flow cytometry data were added in the result section.
- Discussion: Authors wrote :“ … cytokines like IL-1β, IL-6, IL-8, and TNF-α play a crucial role.” It is necessary to clarify the role of different cytokines and cite relevant papers. For instance, TNFa kills cancer cells through the death receptor signaling pathway. However, TNFa can promote cancer metastasis if the death receptor pathway is not functional. The dual mechanisms should be accented.
The role of cytokines was improved in the discussion section, according to suggestions.
Liu, Z.; Zhang, Y.; Zhang, L.; Zhou, T.; Li, Y.; Zhou, G.; Miao, Z.; Shang, M.; He, J.; Ding, N. and Liu, Y. Duality of Interactions Between TGF-β and TNF-α During Tumor Formation. Front Immunol. 2022, 12, 1–14. DOI: 10.3389/fimmu.2021.810286.
Zheng, X.; Turkowski, K.; Mora, J.; Brüne, B.; Seeger, W.; Weigert, A.; Savai, R. Redirecting tumor-associated macrophages to become tumoricidal effectors as a novel strategy for cancer therapy. Oncotarget. 2017, 8, 48436–52. Doi: 10.18632/oncotarget.17061
Shen, Z.; Zhou, R.; Liu, C.; Wang, Y.; Zhan, W.; Shao, Z.; Liu, J.; Zhang, F.; Xu, L.; Zhou, X et al. MicroRNA-105 is involved in the TNF-α-related tumor microenvironment-enhanced colorectal cancer progression article. Cell Death and Disease. 2017, 8, 1–13. DOI: 10.1038/s41419-017-0048-x.
Si, H.; Lu, H.; Yang, X.; Mattox, A.; Jang, M.; Bian, Y.; Sano, E.; Viadiu, H.; Yan, B.; Yau, C.; Ng, S.; Lee, S. K.; Romano, R. A.; Davis, S.; Walker, R. L.; Xiao et al. TNF-α modulates genome-wide redistribution of Np63α/TAp73 and NF-κB c-REL interactive binding on TP53 and AP-1 motifs to promote an oncogenic gene program in squamous cancer. Oncogene, 2016, 35, 44, 5781–5794. DOI: 10.1038/onc.2016.112.
Russo, R. C.; Garcia, C. C.; Teixeira, M. M.; Amaral, F. A. The CXCL8/IL-8 chemokine family and its receptors in inflammatory diseases. Expert Rev of Clin Immunol. 2014, 10, 5, 593–619. DOI: 10.1586/1744666X.2014.894886.
Santangelo, C.; Varì, R.; Scazzocchio, B.; Di Benedetto, R.; Files, C.; Masella, R. Polyphenols, intracellular signaling and inflammation. Annali dell’Istituto Sup Sanita. 2007, 43, 4, 394–405.
Hirahara, K.; Vahedi, G.; Ghoreschi, K.; Yang, X. P.; Nakayamada, S.; Kanno, Y.; O'Shea, J. J.; Laurence, A. Helper T-cell differentiation and plasticity: Insights from epigenetics. Immunol. 2011, 134, 3, 235–245. DOI: 10.1111/j.1365-2567.2011.03483.x.
Hanker, L. C.; Rody, A.; Holtrich, U.; Pusztai, L.; Ruckhaeberle, E.; Liedtke, C.; Ahr, A.; Heinrich, T. M.; Sänger, N.; Becker, S.; Karn T. Prognostic evaluation of the B cell/IL-8 metagene in different intrinsic breast cancer subtypes. Breast Cancer Res Treat. 2013, 137, 2, 407–416. DOI: 10.1007/s10549-012-2356-2.
Deng, F.; Weng, Y.; Li, X.; Wang, T.; Fan, M.; Shi, Q. Overexpression of IL-8 promotes cell migration via PI3K-Akt signaling pathway and EMT in triple-negative breast cancer. Pathol Res Pract. 2020, 223, 1–18. DOI: 10.1016/j.prp.2020.152824.
Todorović-Raković, N.; Milovanović, J. Interleukin-8 in breast cancer progression. J Interf Cytokine Res. 2013, 563–570. DOI: 10.1089/jir.2013.0023.
Ma, Y.; Ren, Y.; Dai, Z. J.; Wu, C. J.; Ji, Y. H.; Xu, J. IL-6, IL-8, and TNF-α levels correlate with disease stage in breast cancer patients. Advanc Clin Exper Med. 2017, 26, 3, 421–426. DOI: 10.17219/acem/62120.
Polat, A.; Tunc, T.; Erdem, G.; Yerebasmaz, N.; Tas. A.; Beken, S.; Basbozkurt, G.; Saldir, M.; Zenciroglu, A.; Yaman, H. Interleukin-8 and Its Receptors in Human Milk from Mothers of Full-Term and Premature Infants. Breast Med. 2016, 11, 5, 247–251. DOI: 10.1089/bfm.2015.0186.
Ustundag, B.; Yilmaz, E.; Dogan, Y.; Akarsu, S.; Canatan, H.; Halifeoglu, I.; Cikim, G.; Denizmen Aygun, A. Levels of cytokines (IL-1β, IL-2, IL-6, IL-8, TNF-α) and trace elements (Zn, Cu) in breast milk from mothers of preterm and term infants. Mediators Inflamm. 2005, 2005, 6, 331–336. DOI: 10.1155/MI.2005.331.
XIE, K. Interleukin-8, and human cancer biology. Cytokine Growth Fact Rev. 2001, 12, 4, 375–391. DOI: 10.1016/S1359-6101(01)00016–8.
Comments on the Quality of English Language
Good English, minor editing is recommended.
R: Thank you for the suggestions, and we apologize for any inconvenience. The text was improved and revised by a native English speaker.

Round 2
Reviewer 1 Report
The authors have addressed all my comments.
Reviewer 2 Report
The authors addressed all my comments properly. The manuscript has been amended and the current version can be accepted.
Minor checking of grammar is required.